# Assessing the Efficacy of Radioactive Iodine Seed Localisation in Targeted Axillary Dissection for Node-Positive Early Breast Cancer Patients Undergoing Neoadjuvant Systemic Therapy: A Systematic Review and Pooled Analysis

**DOI:** 10.3390/diagnostics14111175

**Published:** 2024-06-02

**Authors:** Munaser Alamoodi, Umar Wazir, Janhavi Venkataraman, Reham Almukbel, Kefah Mokbel

**Affiliations:** The London Breast Institute, Princess Grace Hospital, London W1U 5NY, UK; malamoodi@kau.edu.sa (M.A.); umar.wazir@rcsed.ac.uk (U.W.); janhavi.venkataraman@hcahealthcare.co.uk (J.V.); rmokbel7@gmail.com (R.A.)

**Keywords:** neoadjuvant systemic therapy, breast cancer, targeted axillary lymph node, iodine radioactive seed, pathological complete response, staging, metastasis, lymphatic dissemination

## Abstract

Targeted axillary dissection (TAD), employing marked lymph node biopsy (MLNB) alongside sentinel lymph node biopsy (SLNB), is increasingly recognised for its efficacy in reducing false negative rates (FNRs) in node-positive early breast cancer patients receiving neoadjuvant systemic therapy (NST). One such method, ^125^I radioactive seed localisation (RSL), involves implanting a seed into a biopsy-proven lymph node either pre- or post-NST. This systematic review and pooled analysis aimed to assess the performance of RSL in TAD among node-positive patients undergoing NST. Six studies, encompassing 574 TAD procedures, met the inclusion criteria. Results showed a 100% successful deployment rate, with a 97.6% successful localisation rate and a 99.8% retrieval rate. Additionally, there was a 60.0% concordance rate between SLNB and MLNB. The FNR of SLNB alone was significantly higher than it was for MLNB (18.8% versus 5.3%, respectively; *p* = 0.001). Pathological complete response (pCR) was observed in 44% of cases (248/564). On average, the interval from ^125^I seed deployment to surgery was 75.8 days (range: 0–272). These findings underscore the efficacy of RSL in TAD for node-positive patients undergoing NST, enabling precise axillary pCR identification and facilitating the safe omission of axillary lymph node dissection.

## 1. Introduction

The replacement of axillary lymph node dissection (ALND) by the less invasive sentinel lymph node biopsy (SLNB) has become the standard in patients who present with clinically node-negative breast cancer. This shift came after prospective trials showed that there was no oncological compromise in terms of survival [1,2,3,4]. However, the application of this shift to patients presenting as clinically node positive (cN+) who responded favourably to neoadjuvant systemic therapy (NST) was hindered by a lack of evidence regarding oncological safety and high false negative rates (FNRs) associated with SLNB, ranging from 11.9% to 14.2% [5,6,7]. These findings led to the assumption that performing standard SLNB alone was oncologically unsafe.

Targeted axillary dissection (TAD) combines the surgical biopsy following localisation of a marked pathological lymph node (marked lymph node biopsy; MLNB) prior to NST with a SLNB. TAD has emerged as an effective strategy to reduce the FNR of axillary staging in node-positive early breast cancer patients receiving NST [8]. A study from MD Anderson Cancer Centre found the FNR to be 10.1% for SLNB alone, 4.2% when the clipped node alone was evaluated, and 1.4% for SLNB plus evaluation of the clipped node [8]. Notably, the clipped node was not identified as a sentinel node in 23–24% of cases, and preoperative axillary localisation was important to ensure its removal [8]. These findings made it possible to carry out a less invasive procedure, thereby reducing the morbidities associated with ALND, such as lymphedema, restriction of shoulder movement, and the possibility of nerve damage [9]. 

Radioactive ^125^I seed localisation (RSL) is one of the localisation techniques used to mark, identify, and harvest biopsy-proven lymph nodes in patients undergoing NST. Seeds (4.8 mm × 0.8 mm) are made of titanium and contain ^125^I, which emits gamma radiation and has a half-life of sixty days [10,11]. Seeds are maintained under strict conditions at the radionuclide laboratory within the department of nuclear medicine. Seeds are transported in a sterile container or pre-loaded into an 18-gauge spinal needle, which is blocked with specific materials in order to avoid accidental deployment [12]. The standard handheld gamma probes used for sentinel lymph node (SLN) detection can be used to localise both the ^125^I seed as well as radioactive LNs by changing the energy mode. When the seed-containing lymph node has been removed, an intra operative specimen X-ray is performed to confirm seed retrieval. The ^125^I seed must also be disposed of in accordance with radioactive safety guidelines. The maximum amount of time for seed permanence recommended by guidelines in the United States is between 5 and 7 days [12]. The use of RSL in TAD is often referred to as the MARI (marking the axillary lymph node with radioactive iodine seeds) procedure (Figure 1).

Although standard SLNB was considered to be oncologically unsafe, removal of more than one lymph node has been shown to reduce the FNR in a recent meta-analysis [14]. The study showed that studies included in the meta-analysis demonstrated FNRs to be 8% when three or more SLNs were detected and 22% when less than three were detected. The possibility of avoiding ALND when three or more SLNs are retrieved and less than three are positive is supported by a number of studies [5,15].

NST with chemotherapy and targeted therapy has improved the rate of pathological complete response (pCR) attainment. Around 40% of patients achieve pCR in both the breast and axilla, predominantly in certain biologically aggressive early-stage breast cancers, including triple-negative and human epidermal growth factor receptor 2 (HER2)-positive breast cancers [16]. Given the attainment of high pCR levels in this group of patients, the benefit of ALND has been questioned. 

Extensively investigated in numerous studies, RSL’s excellent performance in facilitating surgical excision of non-palpable lesions has been demonstrated. A recent meta-analysis involving 19,820 patients found RSL to be a superior method to the previous standard of care, wire-guided localisation, in terms of surgical efficiency for intraoperative localisation of impalpable breast lesions [17].

This systematic review and pooled analysis aimed to assess ^125^I seeds’ clinical performance during TAD (MLNB plus SLNB) by assessing successful localisation and retrieval rates, concordance between MLNB and SLNB, and the incidence of pCR in clinically node-positive patients undergoing NST.

## 2. Materials and Methods

### 2.1. Literature Search

This study received approval from the multidisciplinary breast cancer board of the London Breast Institute. A comprehensive literature search was conducted using PubMed and Google Scholar databases up to March 2024. The search utilised the following keywords:[radioactive iodine seed] or [radioactive seed localisation (RSL)];[targeted axillary dissection] or [TAD];[breast cancer];[neoadjuvant].

Additionally, bibliographies of relevant studies were examined for potential inclusion. A significance threshold of *p* < 0.05 was applied for statistical analyses.

### 2.2. Inclusion and Exclusion Criteria

Studies identified in the literature search were evaluated based on the following inclusion and exclusion criteria.

#### 2.2.1. Inclusion Criteria

Studies were included if they met the following criteria:Retrospective or prospective cohort design.Investigation of the role of radioactive iodine seed in TAD in patients undergoing neoadjuvant systemic therapy (NST).Availability of data endpoints, including successful localisation and retrieval rate, SLNB-MLNB concordance rate, pathological complete response (pCR), and migration rate.

#### 2.2.2. Exclusion Criteria

Studies meeting the following criteria were excluded:Manuscripts not available in English.Studies involving non-human subjects.Studies with fewer than five eligible cases were excluded to minimise the impact of early learning curve experience.Non-peer-reviewed studies.Case reports, reviews, and trial updates.

## 3. Results

### 3.1. Literature Search Results 

The search yielded 98 articles, of which 6 met the inclusion criteria, encompassing 574 patients (Table 1; Figure 2) [13,18,19,20,21,22].

### 3.2. Subsection

Six studies, involving 574 patients, met the inclusion criteria. The pooled average age was 51.9 years (range: 22–82). The pooled analysis revealed the following:Successful localisation rate: 97.6% (560/574) [95% confidence interval (CI), 0.97–0.98].

This is demonstrated in the Forest plot shown in Figure 3.

Retrieval rate: 99.8% (573/574).Concordance rate between SLNBs and MLNBs: 60.0% (289/481) [95% CI, 0.56–0.64]. Subgroup analysis of studies reporting the pathological status of MLNBs and SLNBs separately revealed a FNR of 5.2% for MLNBs and 18.8% for SLNBs. Chi squared equaled 18.398 with 1 degree of freedom. The two-tailed *p* value was less than 0.0001.pCR was observed in 44% of cases (248/564) [95% CI, 0.35–0.45], with no reported migration or procedure-specific complications.In one study, the ^125^I was not retrieved, and was found in fibrosed tissue during pathological assessment. This was thought to be due to severe regression of the node in response to NST [16]. In another study, the seed was retrieved inferiorly in the axilla [20].The successful deployment rate was 100%, but one patient required repeat deployment due to seed misplacement during ultrasound-guided localization [18].Localisation was compromised in one patient due to the inability to visualise the clip by ultrasound, which led to subjecting the patient to ALND [18].The pooled average number of lymph nodes retrieved during the TAD procedure was 2 (range: 1–11).The pooled average interval duration from magnetic seed deployment to surgery was 75 days (range: 0–272 days).

## 4. Discussion

### 4.1. Performance of ^125^I Seed in TAD 

Our pooled analysis spanning 574 procedures provides strong evidence supporting the efficacy of using ^125^I seeds in TAD, with successful deployment, localisation, and retrieval rates of 100%, 97.6%, and 99.8%, respectively. Furthermore, we observed a concordance rate of 60.0% between MLNB and SLNB. The FNR for SLNB alone (18.8%) was significantly higher than the acceptable threshold, and was consistent with those reported in previous studies [23]. Furthermore, it was significantly higher than that of MLNB (5.3%), underscoring the importance of incorporating MLNB in staging the axilla post-NST in patients presenting with node-positive disease. This was emphasised in the updated National Comprehensive Cancer Network (NCCN) 2022 guidelines.

The use of ^125^I seed has several advantages, including patient comfort, increased ease of scheduling with the decoupling of surgery and radiology procedures, and decreased risk of displacement. However, the nature of its radioactivity limits its use in centres not equipped to deal with the strict safety procedures, as well as limiting the number of seeds used in a patient. Deployment of localising seeds pre-NST has the advantage of accurate and easier localisation. This is due to the difficulty of identifying a treated LN that regresses in size. The strict time frame in which ^125^I seeds have to be deployed before removal in certain jurisdictions (up to 5 days in the USA) also limits its use and restricts it to the two-stage approach, which is less accurate and more expensive. Furthermore, the mean cost of ^125^I seed-assisted TAD was found to be 25% superior to the mean cost of ALND, and the mean total cost of the hospital stay for TAD was 20% superior to the mean cost of ALND. The authors reported that the mean cost of TAD was similar to the mean cost of both ALND and SLNB performed during the same procedure. Despite increased procedural costs, with a lesser impact on total hospital stay costs TAD was beneficial for 50% of patients. These patients avoided the unnecessary morbidity associated with ALND [20]. It is also worth noting that non-procedural costs, which have an impact on mid-term and long-term expenditures, can be minimised as a result of TAD success. Such costs can include the omission of radiotherapy and ALND in node-positive patients who achieve pCR in the TAD nodes [20].

### 4.2. Comparison of Wireless Technologies for Localisation 

The ^125^I seed, due to its radioactive nature and signal decay, has limitations as described above. However, it has the advantage of being cost-effective. The advent of new non-radioactive technologies has overcome the limitations of ^125^I seed localisation. These technologies include magnetic seed (Magseed^®^, Camnbridge, UK), SAVI SCOUT^®^ (Aliso Viejo, CA, USA) or radar reflector localisation (RRL), and LOCalizer^®^ (Marlborough, MA, USA) radio-frequency identification (RFID) tags. A Magseed is a 5 mm stainless steel paramagnetic seed that uses magnetic fields [24]. A probe induces and detects the magnetic field of a Magseed with an audio signal up to a depth of 4 cm [25]. Magseed has the advantage of being deployed at the time of lymph node biopsy with a high localisation rate (99.86%) [26]. Its interference with MRI and the necessity of removing all metallic equipment from the surgical field before localisation are significant drawbacks [27].

We recently conducted a systematic review of the literature regarding the performance of magnetic seeds in TAD (in press). The review included 494 patients and 497 procedures, and demonstrated a 100% successful deployment rate, a 94.2% localisation rate, a 98.8% retrieval rate, and a 68.8% concordance rate. pCR was observed in 47.9% of cases, and the mean duration of implantation was 37 days (range: 0–188).

In contrast, the SAVI SCOUT reflector, which utilises micro-impulse infrared radar [28], does not interfere with MRI and offers precise distance information between the reflector and the handheld probe. This differs from the current version of the Magseed device, which only provides an audio signal. Although the SAVI SCOUT is larger than a Magseed or a radioactive iodine seed, once deployed, its position cannot be adjusted [28]. However, the larger size of the SAVI SCOUT reflector facilitates visualisation on ultrasound and simplifies identification during localisation [29].

Our recent analysis involving 252 TAD procedures has shown the efficacy of the SAVI SCOUT localisation in aiding TAD, with a 100% successful deployment rate, a 99.6% successful localisation rate, a 100% retrieval rate, and an 81% concordance rate between SLNB and MLNB [29]. The average interval from reflector deployment to surgery was 52 days (range: 1–202), and pCR was observed in 42% (95% CI: 36–48) of cases.

The LOCalizer utilises a 12 mm radio-frequency identification (RFID) ‘tag’ and and radio wave signalling [30]. It can be deployed at the time of biopsy and can transmit an audio–visual reading like the SAVI SCOUT [31]. Unlike the SAVI SCOUT, it interferes with MRI, though to a lesser extent than Magseed, and is susceptible to migration and malpositioning [32]. However, there are currently limited data on the performance of RFID tags in TAD, with only 40 procedures reported, showing a 2.5% failed localization rate [31]. The wide bore of the introducer needle and the glass casing of the radio-frequency tag represent inherent limitations for the LOCalizer.

Thus, all three wire-free technologies exhibit excellent efficacy in TAD. Minimal MRI artefacts and the absence of radioactivity make the SAVI SCOUT the preferred method for TAD localisation. [29].

### 4.3. Disadvantages of RSL 

Despite the excellent performance of radioactive iodine seeds in breast [17] and axillary surgery, as demonstrated in this review, it is important to highlight that this approach has certain limitations and disadvantages. Radioactive materials in the medical setting attract government-mandated regulations necessitating compliance with prescribed procedures surrounding storage, handling, disposal, specific training of personnel, and specific documentation required by regulatory authorities. Areas need to be clearly marked and infograms may need to be prominently displayed. Accounting for each seed is paramount from a health and safety perspective. All this has implications for the cost of deploying such a service [33].

In certain jurisdictions, the duration of implantation is restricted to 5 days due to concerns about the radiation absorbed by the patient, as well as diminishing radioactivity over the sixty-day half-life of the seed [33].

Furthermore, seeds may be difficult to locate by ultrasound, owing to their size. Seeds are typically placed under ultrasound or mammography guidance. MRI guidance is not advised due to the risk of losing a seed and the ability to locate it using a hand-held Geiger counter. After placement, radioactive seeds may cause minimal susceptibility artefacts on MRI scans, similar to those observed around clips or coils. Migration of implanted seeds is rare, with reported average migration distances of 0.9 mm [34]. Finally, it is theoretically feasible that the low radiation energy of a radioactive seed can eradicate minimal residual disease [35].

### 4.4. Oncological Safety of TAD 

Global practices regarding axillary staging post-NST for initially node-positive disease vary, influenced by tumour biology. There is a growing body of evidence supporting the reduction of axillary treatment in patients who respond well to NST. Chun et al. [36] demonstrated no survival difference between SLNB and ALND post-NST, establishing SLNB as the preferred option for patients achieving pCR. Five-year data from NSABP B-51 confirmed the safety of axillary treatment de-escalation. A meta-analysis by Rana et al. at the 2023 San Antonio Breast Cancer Symposium supported these findings [37]. In contrast, Galimberti et al. reported on a retrospective study regarding outcomes of patients who underwent SLNB after achieving pCR, and found that performing SLNB alone after pCR did not result in worse survival outcomes [38].

The SenTa study compared outcomes in patients undergoing TAD alone versus TAD with ALND, and suggested similar recurrence rates for TAD alone in patients with good NST responses. However, the study’s observational design and limited follow-up are notable limitations [39]. The suitability of TAD in extensive pre-NST axillary disease remains uncertain, although it shows low false-negative and locoregional failure rates in the medium term. Longer follow-up and updated guidelines are needed as TAD becomes standard practice. Wu et al. [40] reported similar outcomes between TAD alone and TAD with ALND, further supporting the safety of TAD. Schlafstein et al. found no survival benefit from regional nodal irradiation (RNI) in patients converting from cN1 to ypN0 status following NST [41]. 

Five-year data from NSABP B-51 confirmed the safety of axillary treatment de-escalation. The NRG Oncology/NSABP B-51/RTOG 1304 study supports omitting RNI in patients transitioning from cN1 to ypN0 status based on SLNB after NST [1]. The ongoing non- inferiority randomised TAXIS trial is evaluating tailored axillary surgery (TAS) followed by ALND and RNI excluding the dissected axilla or RNI including the full axilla in patients with clinically node positive disease, targeting disease-free survival and quality of life as endpoints [42].

Nijveldt et al. recently assessed the success of TAD using ^125^I seed localisation [43]. Their treatment algorithm for adjuvant therapy was based on the number of suspected axillary lymph nodes pre-NST and, subsequently, the response of the TAD node(s). Post-surgery patients had no further axillary treatment recommended if there were one to three suspected positive axillary lymph nodes pre-NST and a pCR of the TAD node(s). Axillary radiotherapy was recommended if there were between one to three suspected positive axillary lymph nodes pre-NST and a tumour-positive TAD node(s), or in a case of more than three suspected positive axillary lymph nodes, pre-NST and a pCR of TAD node(s). An ALND with axillary radiotherapy was only recommended if there were more than three suspected positive axillary lymph nodes pre-NST, and TAD node(s) were positive for malignancy. A total of 312 TAD procedures were successfully performed in 309 patients. In 134 (43%) cases, pCR of TAD lymph nodes was observed. Per treatment protocol, 43 cases (14%) did not receive any axillary treatment, 218 cases (70%) received adjuvant axillary radiotherapy, and 51 cases (16%) underwent an ALND. During a median follow-up of 2.8 years, 46 patients (14%) developed recurrence, of whom 11 patients (3.5%) had axillary recurrence. The authors concluded that the introduction of the TAD procedure resulted in a reduction of 84% of previously indicated ALNDs. Moreover, 18% of cases did not receive adjuvant axillary radiotherapy. These data show that implementation of de-escalation of axillary treatment with the TAD procedure appear to be successful. In another study, TAD, assisted by ^125^I seed localisation of lymph nodes, prevented ALND in 80% of cN+ patients with a three-year axillary recurrence-free rate of 98% [44].

### 4.5. Limitations 

This review marks the first comprehensive analysis of all existing studies regarding the effectiveness of RSL during TAD, involving more than 400 patients. However, the analysed studies varied in their approaches, such as the timing of seed deployment, monitoring response to NST, criteria for TAD selection, and lymph node retrieval numbers. Three [18,20,21] of the six studies included in the analysis were retrospective in nature and prone to misclassification and selection bias.

Furthermore, three studies [20,21,22] included a small sample size of less than 50. None of the studies analysed included direct comparisons between RSL–TAD and alternative localisation methods, and there was a lack of data on oncological outcomes across all studies. 

Furthermore, performance of RSL–TAD in BC patients with high initial lymph node involvement (≥ three clinically suspicious LNs) was not addressed in our review due to lack of data. Patients with high lymph node involvement are often excluded from larger studies of TAD or other axillary surgical approaches. Therefore, assessing the FNR of TAD compared to ALND in patients with ≥ 3 clinically positive LNs in a larger cohort is necessary. Extensive initial LN involvement increases the likelihood of a false-negative TAD result, potentially leaving involved LNs in the axilla if only TAD is performed.

These limitations highlight areas where the literature may benefit from further investigation, posing potential clinical questions for future research.

## 5. Conclusions

The ^125^I seed demonstrates high reliability and accuracy in TAD, providing a robust method for localising pathological axillary lymph nodes in individuals undergoing NST for early-stage breast cancer. This technique facilitates the safe reduction of axillary surgical interventions, thereby reducing morbidity rates and improving quality of life. However, the complex radiation safety regulations required present a significant limitation to its widespread adoption.

## Figures and Tables

**Figure 1 diagnostics-14-01175-f001:**
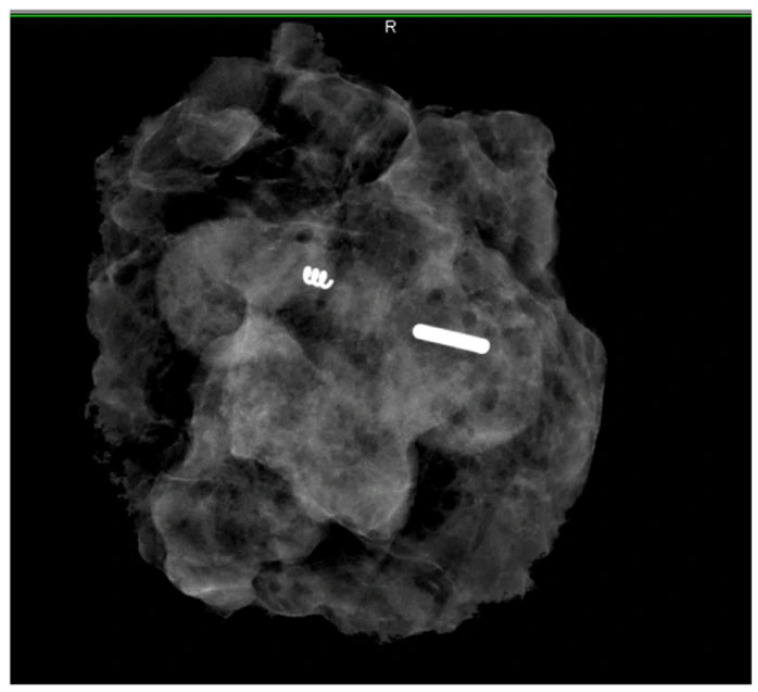
A specimen radiograph demonstrating the ^125^I seed in a specimen in the lymph node in addition to the marker coil. Reproduced from Zatecky et al. [13].

**Figure 2 diagnostics-14-01175-f002:**
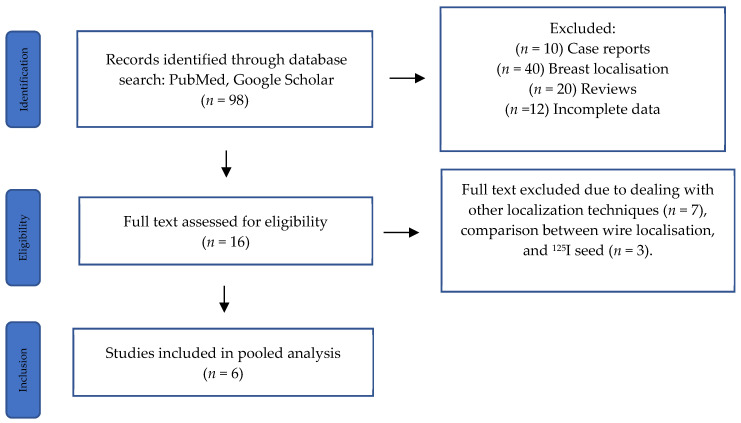
PRISMA flow diagram illustrating the inclusion and exclusion of studies.

**Figure 3 diagnostics-14-01175-f003:**
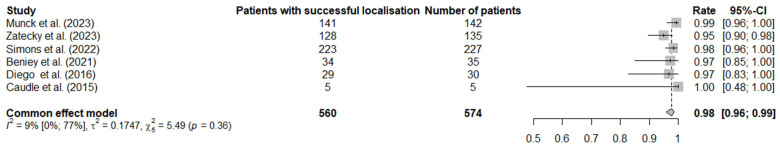
Forest plot demonstrating the pooled successful localisation rate [13,18,19,20,21,22].

**Table 1 diagnostics-14-01175-t001:** Pooled analysis of included studies. NST: neoadjuvant systemic therapy, pCR: pathological complete response, SLNB: sentinel lymph node biopsy, MLNB: marked lymph node biopsy, FN: false negative.

Study	Citation	Number of Patients Pre- or Post-NST	Mean Age in Years	pCR (%)	Retrieval Rate	Localization Success Rate	Migration Rate	Mean Implantation Duration (Days)	Median Number of Nodes Harvested	SLNB–MLNB Concordance Rate	FN MLNB	FN SLNB
Zatecky et al. (2023)	[13]	142	51 (26–82)	58/142 (40.8%)	142/142 (100%)	141/142 (99.3%)	0	146.5 (101–272)	2 (0–7)	94/130 (72.3%)	6/84	18/84
Munck et al. (2023)	[18]	135	49.4 (26–80)	84/135 (62.2%)	135/135 (100%)	128/135 (94.8%)	4/135	0	3.2 (1–10)	35/128 (27.3%)	0/51	No data
Simons et al. (2022)	[19]	227	52 (22–77)	70/223 (31.4%)	227/227 (100%)	223/227 (98.2%)	0		2 (1–8)	134/188 (71.3%)	10/155	22/129
Beniey et al. (2021)	[20]	35	49 (29–76)	17/34 (50%)	34/35 (97.1%)	34/35 (97.1%)	1/35	0 (day of surgery)	-	-	-	-
Diego et al. (2016)	[21]	30	55 (30–71)	19/30 (63.3%)	30/30 (100%)	29/30 (96.7%)	0		4 (1–11)	22/30 (73.3%)	0/11	-
Caudle et al. (2015)	[22]	5	55 (35–69)		5/5 (100%)	5/5 100%	0	5 (0–5)	2.4 (1–6)	4/5 (80%)	-	-
Total		574	51.9 (22–82)	248/564 (44%)	573/574 (99.8%)	560/574 (97.6%)	5/170 (2.9%)	75.8 (0–272)	2 (0–11)	289/481 (60.0%)	16/301 (5.3%)	40/213 (18.8%)

## Data Availability

Datasets generated during this study are publicly available in this open access publication without any restrictions.

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
