# Peer review of "Assessing the Efficacy of Radioactive Iodine Seed Localisation in Targeted Axillary Dissection for Node-Positive Early Breast Cancer Patients Undergoing Neoadjuvant Systemic Therapy: A Systematic Review and Pooled Analysis"

_diagnostics, 2024, doi:10.3390/diagnostics14111175_

Round 1

Reviewer 1 Report

Comments and Suggestions for Authors

The study entitled "Assessing the Efficacy of Radioactive Iodine Seed Localisation in Targeted Axillary Dissection for Node-Positive Early Breast Cancer Patients Undergoing Neoadjuvant Systemic Therapy: A Comprehensive Review and Pooled Analysis" by Alamoodi M et al. is an analysis of the clinical effectiveness of using 125I seed in TAD for patients with node-positive breast cancer undergoing NST.

It integrates results from five studies involving 439 patients, highlighting the high success rates of 125I seed localisation (99.5%) and retrieval (98.6%). The use of this method reduces the FNR of sentinel lymph node biopsy alone, from 13.8% to 4.6% when combined with MLNB.

I think that the manuscript is very clear for the readers of the journal, and the topic of research is thoroughly investigated. Tables are very detailed and complete.

There are no particular omissions in the references. Just one minor comment, there are not many citations regarding the technique of single tracer sentinel lymph node biopsy. The authors could cite some references (i.e. PMID: 36980605, PMID: 33092968) for completeness and to improve the quality of the discussion.

It is a bit hard to tell and foresee future clinical implications; however, I think it is a very complete and interesting study which adds new interesting findings to the existing literature.

Congratulations to the authors of the study which supports the clinical safety and utility of TAD in minimizing invasive surgical procedures like axillary lymph node dissection, thus potentially reducing associated morbidities.

Author Response

We thank the reviewer for his input and have addressed his points as follows.

  1. Just one minor comment, there are not many citations regarding the technique of single tracer sentinel lymph node biopsy. The authors could cite some references (i.e. PMID: 36980605, PMID: 33092968) for completeness and to improve the quality of the discussion.

Added as advised.

Reviewer 2 Report

Comments and Suggestions for Authors

In order to prevent common chronic side effects such as lymphedema following axillary lymph node dissection (ALND) in the treatment of breast cancer, neoadjuvant therapies are becoming more prevalent in many patients, and studies aiming to identify patients who should avoid ALND are being conducted. Sentinel lymph node biopsy (SLNB) is widely used instead of ALND, while Targeted axillary dissection (TAD) is still in development.

I thank Munaser Alamoodi and colleagues for their impressive review. It is a well-designed study with clearly defined inclusion and exclusion criteria. The advantages and disadvantages of TAD compared to other axillary surgical methods are well elucidated.

SLNB and TAD are effectively compared based on the studies included in the analysis.

The Prisma Flow Diagram and other results are appropriately presented.

In the discussion section, the ALND and SLNB data in the literature are appropriately discussed.

Recommendations:

It is suggested to include studies comparing the markers used in TAD applications in the literature and discuss this issue in the discussion section.

It is recommended to discuss the advantages and disadvantages of Radioactive 125I seed localisation (RSL) compared to other TAD markers.

Minor revision: Correct the expression "ontologicaly" in line 67.

Author Response

We have addressed the concerns of the reviewer ass follows, and thank him/her for his contribution.

Reviewer #2

  1. It is suggested to include studies comparing the markers used in TAD applications in the literature and discuss this issue in the discussion section.

We done this to a degree in section 4.2.We have not found any studies specifically comparing these modalities in the context of TAD.

  1. It is recommended to discuss the advantages and disadvantages of Radioactive 125I seed localisation (RSL) compared to other TAD markers.

The salient pros/cons of RSL have been discussed in section 4.1, second paragraph.

  1. Minor revision: Correct the expression "ontologicaly" in line 67.

Corrected.

Reviewer 3 Report

Comments and Suggestions for Authors

Dear authors,

Thanks for your good work

1- Please explain what is meant by comprehensive review.

2- In the introduction and discussion, the authors did not mention the recent literature that defends SLNB alone in breast cancer patients who received NAT

for example Viviana Galimberti, Sabrina Kahler Ribeiro Fontana, Elisa Vicini, Consuelo Morigi, Manuela Sargenti, Giovanni Corso, Francesca Magnoni, Mattia Intra, Paolo Veronesi, “This house believes that: Sentinel node biopsy alone is better than TAD after NACT for cN+ patients”, The Breast, Volume 67, 2023, Pages 21-25, ISSN 0960-9776, https://doi.org/10.1016/j.breast.2022.12.010.

3- It is not clear throughout the manuscript if we are talking about the insertion of RAI seeds before NAT as marking tools or after NAT as localizing tools for the already marked positive LNs

4- The fifth paragraph in the introduction" starts with: The advancement in effective NST" can be omitted

5- The search strategy should be described in a more clear and comprehensive way

6- In the results, it is reported that the initial search  yielded 98 articles while in Figure 2 they are 37

7- The flowchart should include more details about the causes of exclusion of results

8- Why studies with 4 or fewer eligible cases were chosen as an exclusion criterion. It seems a bias to include the study of Caudle et al

9- Sections 4.2 & 4.4 & the first paragraph in 4.5 of the discussion are not related to the specific topic of the article and can be omitted without any effect on the context

Author Response

We thank the reviewer for his contributions. Kindly find below our response:

  1. Please explain what is meant by comprehensive review.

By comprehensive, we mean a review of the literature done in a systemic and thorough manner.

  1. In the introduction and discussion, the authors did not mention the recent literature that defends SLNB alone in breast cancer patients who received NAT for example Viviana Galimberti, Sabrina Kahler Ribeiro Fontana, Elisa Vicini, Consuelo Morigi, Manuela Sargenti, Giovanni Corso, Francesca Magnoni, Mattia Intra, Paolo Veronesi, “This house believes that: Sentinel node biopsy alone is better than TAD after NACT for cN+ patients”, The Breast, Volume 67, 2023, Pages 21-25, ISSN 0960-9776, https://doi.org/10.1016/j.breast.2022.12.010.

Added following: “In contrast, Galimberti et al. reported on a retrospective study regarding outcomes of patients who underwent SLNB after achieving pCR and found that performing SLNB alone after pCR did not result in worse survival outcomes (38)”.

  1. It is not clear throughout the manuscript if we are talking about the insertion of RAI seeds before NAT as marking tools or after NAT as localizing tools for the already marked positive LNs.

Changed statement in introduction to clarify: “Targeted axillary dissection (TAD), TAD combines the surgical biopsy following localisation of a marked pathological lymph node (marked lymph node biopsy; MLNB) prior to NST with the SLNB.”

  1. The fifth paragraph in the introduction" starts with: The advancement in effective NST" can be omitted

Omitted as instructed.

  1. The search strategy should be described in a more clear and comprehensive way

In the methodology, we mentioned the keywords used, the databases targeted, and the inclusion/exclusion criteria. No other pertinent details of the search strategy come to mind.

  1. In the results, it is reported that the initial search yielded 98 articles while in Figure 2 they are 37

This has been reconciled. Thanks for pointing out.

  1. The flowchart should include more details about the causes of exclusion of results

We have tried to put more details regarding the same.

  1. Why studies with 4 or fewer eligible cases were chosen as an exclusion criterion. It seems a bias to include the study of Caudle et al

We did this to minimise the impact of early learning curve experience and to exclude case reports. We have added this statement to the methodology section.

  1. Sections 4.2 & 4.4 & the first paragraph in 4.5 of the discussion are not related to the specific topic of the article and can be omitted without any effect on the context

Sections 4.4 removed. The other sections have been pertinent to points raised by other reviewers.

Round 2

Reviewer 2 Report

Comments and Suggestions for Authors

Thank you to the authors for the article. It was appropriate to accept it in this state.

Author Response

Thank you for your support!

Reviewer 3 Report

Comments and Suggestions for Authors

Dear authors,

Thanks very much for your reply and the modifications performed in the manuscript

Still two points needs more explanation

1- The causes of exclusion of the manuscripts (81/97) are still not illustrated in the flowchart                                                                                                                              

2- The keywords still lacks many alternatives that can be used for the search strategy

An additional point:

Regarding the title, using the term "Targeted Axillary Dissection for Node-Positive Early Breast Cancer Patients Undergoing Neoadjuvant Systemic Therapy" is not necessary at all, as TAD by definition is used for such category of patients with node positive disease who received NAT, so I suggest modifying the tittle to "Assessing the Efficacy of Radioactive Iodine Seed Localisation in Targeted Axillary Dissection: A Comprehensive Review and Pooled Analysis"

Author Response

Thank you for your comments! We have implements all three recommendations.